# Long-Term Survival and Factors Associated with Increased Mortality in Patients with Ocular Adnexal Lymphomas

**DOI:** 10.3390/cancers16122252

**Published:** 2024-06-18

**Authors:** Diego Strianese, Maria Paola Laezza, Fabio Tortora, Giancarlo Fusco, Oreste de Divitiis, Antonella D’Aponte, Francesco Briganti, Andrea Elefante

**Affiliations:** 1Department of Neuroscience, University of Naples Federico II, 80131 Naples, Italy; strianes@unina.it (D.S.); m.laezza@studenti.unimol.it (M.P.L.); tudapont@unina.it (A.D.); 2Department of Medicine and Health Sciences, University of Molise, 86100 Campobasso, Italy; 3Department of Advanced Biomedical Sciences, University of Naples Federico II, 80131 Naples, Italy; fabio.tortora@unina.it (F.T.); gianc.fusco@studenti.unina.it (G.F.); frabrig@unina.it (F.B.); 4Department of Neurosciences and Reproductive and Odontostomatological Sciences, University of Naples Federico II, 80131 Naples, Italy; oreste.dedivitiis@unina.it

**Keywords:** lymphoma, diagnostic radiology, prognosis, mortality

## Abstract

**Simple Summary:**

Orbital and ocular adnexal lymphoma (OAL) refers to a type of cancer affecting the eye and its surrounding structures. It typically originates from lymphoid tissue and can manifest with different variants, including the non-Hodgkin subtype. While prognosis is generally favorable, this relies on different factors, including early detection, prompt treatment, and cancer subtype. This observational study on diagnostic and prognostic factors in OAL aims to provide significant information about the relationship between imaging and disease outcomes. We conclude that by analyzing radiological features such as subtype, fluorodeoxyglucose avidity, and bone marrow involvement, researchers can identify potential prognostic indicators, ultimately helping clinicians in predicting disease progression, refining diagnostic protocols, and improving patient outcomes.

**Abstract:**

Orbital and ocular adnexal lymphoma (OAL) affects the orbit and the surrounding structures and can arise as several subtypes with variable prognoses. We performed an observational study on the relationship between OAL subtype, diagnostic features, and prognosis to offer valuable insights into imaging techniques, such as Positron Emission Tomography with 2-deoxy-2-[fluorine-18] fluoro-D-glucose integrated with Computed Tomography (^18^F-FDG PET-CT), in predicting outcomes. With this aim, we retrospectively reviewed 99 patients with OALs, recording demographics, cancer subtype, location and treatment, ^18^FDG avidity, and bone marrow positivity. We divided patients into Group 1 (those presenting with extranodal marginal zone lymphoma—EMZL) and Group 2, including all other subtypes. The primary outcome was long-term cancer-specific survival (CSS) based on key predictors, performed through Kaplan–Meier curves and the log-rank test, with a *p* < 0.05 significance threshold. The mean patient age was 67 years (57–75.5). The most frequent histopathologic subtypes were EMZL lymphoma in 69 patients (69.7%), small lymphocytic lymphoma (11.1%) and diffuse-large B-cell lymphoma (10.1%). Patients of Group 1 showed a better prognosis (CSS = 80%) compared to those of Group 2 (CSS = 60%) (*p* = 0.01). In patients with high-grade lymphoma, the occurrence of ^18^FDG avidity (*p* = 0.003) and bone marrow positivity (*p* = 0.005) were related to a worse prognosis. In our group, EMZL was the most prominent subtype of OALs and exhibited the best prognosis, low 18FDG avidity, and bone marrow negativity. By observing specific patterns in radiological findings, it is possible to increase our understanding of disease progression, treatment response, and the overall prognosis in OAL patients.

## 1. Introduction

Lymphomas are malignant lymphoid tumors arising from the clonal proliferation of either B-lymphocytes, T-lymphocytes, or, less commonly, natural killer (NK) cells. Lymphomas are divided into two major categories, namely Hodgkin lymphoma (HL) and non-HL (NHL). While HL usually has a B-lymphocyte origin, NHL is a more heterogeneous group of lymphomas of different clonal origins, containing B-cell lymphoma, which constitute more than 85% of lymphoid neoplasms [1]. When originating in an extranodal site, these lesions can arise in the orbit, with ocular adnexal lymphoma (OAL) therefore referring to a malignant lymphoproliferative disorder originating from orbital tissue and the lacrimal gland, as well as the eyelids and conjunctiva. Although OAL accounts for the majority of orbital malignancies in adults, it is still a rare disease, accounting for less than 1% of all NHLs [2,3]. The majority of OALs are non-Hodgkin B-cell lymphomas and can be observed more commonly in adults in the seventh decade of life. These lesions are mainly unilateral, with bilateral OALs found in 7–24% of cases [4]. Extranodal marginal zone lymphoma of mucosa-associated lymphoid tissue (MALT lymphoma) is the most common subtype, followed by diffuse large B-cell lymphoma, follicular lymphoma, mantle cell lymphoma, and lymphoplasmacytic lymphoma [5,6].

Different studies identified negative prognostic factors for OAL, such as older age (e.g., greater than 60 years), node involvement, and elevated serum lactate dehydrogenase levels [7]. Several reports have also documented a relative association between the occurrence of these lesions and some infections, such as Hepatitis C virus (HCV) and *Chlamydia psittaci* [8,9,10], with an incidence that also seems to have increased in the last two decades [11]. As OAL can arise in the conjunctiva, eyelids, and orbit, including the lacrimal gland [12], and as it accounts for the majority of orbital malignancies in adults, as mentioned before, the correct identification of clinical and radiological features, as well as the prognosis of these patients, is mandatory. The purpose of this study was therefore to describe clinical features, prognostic outcomes, and long-term (more than 10 years) survival in a cohort of patients with OAL.

## 2. Materials and Methods

This study was a large retrospective single-center study of OAL. The medical records of 99 patients with a histologic diagnosis of B-cell OAL involving the orbital adnexal region, obtained from 1 January 2000 through 31 December 2015 at the Orbital Unit of Federico II of Naples, were identified and included in this study.

The clinical data collected included age at diagnosis, sex, histological subtype, tumor location, rate of survival, outcome, surgical procedure, radiation therapy, chemotherapy, and basic ophthalmological parameters [13].

Inclusion criteria were the following: (1) patients who received a diagnosis of orbital lymphoma (2) and were older than 18 years old, with (3) an available medical record related to their diagnostic work-up. Patients whose (1) clinical data were incomplete or with (2) a poor preservation of specimens were excluded from this study.

In all subjects, a complete ophthalmic examination was performed, and it included the best corrected visual acuity, color vision testing, and inspection and palpation of the eyelids and orbit, as well as an ocular motility evaluation, intraocular pressure measurement, and ophthalmoscopy. All patients underwent diagnostic incisional biopsy, and all specimens were stained with hematoxylin and eosin and analyzed immunohistochemically for histopathologic examination. A complete diagnostic workup of OAL, including Computed Tomography (CT) and/or Magnetic Resonance Imaging (MRI) of the orbital area, full-body Positron Emission Tomography with 2-deoxy-2-[fluorine-18] fluoro-D-glucose integrated with CT (^18^F-FDG PET-CT), and bone marrow biopsy was performed [14]. PET-CT studies were performed at our hospital using the same machine, with results that were blinded and reviewed by a nuclear medicine specialist. The PET-CT results were reported using two methods of evaluation, qualitative and semi quantitative, via visual assessment and maximal standardized uptake value (SUVmax) record, respectively. In particular, visual assessment was used as a dichotomous measure (i.e., either pathological FDG uptake or no uptake). Data on systemic involvement were collected according to the eighth edition of the American Joint Committee on Cancer (AJCC) TNM classification system. The patients were divided, based on their histopathologic subtype, into the following 6 groups: extranodal marginal zone B-cell lymphoma (EMZL), small lymphocytic lymphoma (SLL), diffuse large B-cell lymphoma (DLBCL), mantle cell lymphoma (MCL), follicular lymphoma (FL) and lymphoplasmacytic lymphoma (LPL). As the primary outcome of this study was long-term cancer-specific survival (CSS), survival curves were constructed using the Kaplan–Meier method to account for censored survival times and were compared with the log-rank test. A *p*-value < 0.05 was considered statistically significant. Associations between categorical variables, such as full-body positron emission tomography–computed tomography and bone marrow biopsy, were analyzed using the chi-square test.

## 3. Results

### 3.1. Clinical Features

Ninety-nine patients affected by B-cell OAL were included in this study. The main clinical and demographic data and each histopathologic subtype are summarized in Table 1.

The main clinical sign observed was proptosis, followed by the evidence of a mass in the orbit or eyelid, along with swelling and globe displacement. The majority of OALs were EMZLs (69.7%, *n* = 69). In this group, the median age was 66 years, the disease was mainly unilateral (95.6%, *n* = 66), and a T2N0M0 stage was present in 53.6% of cases. Fifteen patients (21.7%) showed several risk factors, such as a diagnosis of HCV infection, which was an occurrence observed in six patients.

Eleven patients (11.1%) were diagnosed with small lymphocytic lymphoma (SLL), also presenting with a median age of 66 years. The disease was unilateral in 10 cases, primary in almost all the cases, and only a few patients (18%, *n* = 2) had risk factors. The majority of patients had a T2N0M0 (63.6%, *n* = 7) stage at diagnosis.

Ten patients (10.1%) were diagnosed with a DLBCL. The median age of the group was 64.5 years, with a higher male prevalence (70%, *n* = 7). The disease was unilateral in all cases and staged as T2N0M0 in 40% of cases (*n* = 4).

Finally, two patients (2%) received a diagnosis of FL, with a median age of 75.5 years and a disease unilateral and primary in all cases, while just one case was diagnosed with an LL.

The remaining six patients were diagnosed with MCL, with a median age of this last group being 73 years, showing a higher male prevalence (83.3%, *n* = 5). The disease was unilateral in four cases and bilateral in two cases and staged as T2N0M0 in 50% of cases (*n* = 3).

The median follow-up was 90 months. Relapse was observed in 17 patients (17.2%). Among them, twelve patients (70.5%) had a diagnosis of EMZL. One relapsed case (5.9%) belonged to the DLBCL subtype. One (5.9%) had a diagnosis of SLL, and two patients (11.8%) belonged to the MCL subtype. The disease was widespread and treated with combined therapy (chemotherapy and external beam radiation therapy—EBRT) in all cases.

All lymphoma long-term CSS was 80%, as shown in Figure 1.

The group with marginal lymphoma (Group 1) showed a higher survival rate compared to Group 2 (*p* = 0.01) (Figure 2).

Moreover, all enrolled patients with PET-CT and bone marrow positivity presented a lower survival rate compared to those with bone marrow negativity (*p* = 0.003) (*p* = 0.005) (Figure 3 and Figure 4).

A graphical representation of MRI findings in a patient with a typical orbital lymphoma is available in Figure 5.

### 3.2. Treatment

Of the 69 primary EMZLs, 44 (63.8%) were treated with EBRT, and 7 (10.1%) were treated with chemotherapy using the CHOP (Cyclophosphamide, Hydroxydaunorubicin, Vincristine and Prednisone) regimen. Twelve patients (17.4%) were treated with a combination regimen of chemotherapy and EBRT. A combination regimen of chemotherapy and immunotherapy (CHOP and Rituximab) was used in only five cases (7.2%) of primary EMZL. In one case (1.4%), the patient refused any intervention and chose to undergo regular controls. Among the eleven patients with primary SLL, seven received EBRT, a combination regimen was used in three patients (CHOP and EBRT), and in one case, the patient refused any intervention and chose to undergo regular controls. The only patient with LL refused any intervention.

Among the ten primary DLBCLs, four (40%) were treated with EBRT, one (10%) was treated with chemotherapy, three (30%) were treated with a combination regimen (CHOP and EBRT), and the remaining two (20%) patients refused any intervention and chose to undergo regular controls. Among the six patients with primary MCL, three underwent EBRT (50%), one patient (16.6%) was treated with EBRT in combination with chemotherapy, and two (33.3%) were treated with chemotherapy using the CHOP regimen. All cases of FL were treated with chemotherapy (Table 2).

Regarding Radiotherapy (RTx), 77 patients (77.8%) underwent radiation to the entire involved site, such as the entire conjunctiva/eyelid or entire orbit, with a median total and fractional dose of 25.2 Gy (range, 20–30.6 Gy) and 2.0 Gy (range, 1.8–2.2 Gy), respectively. 

## 4. Discussion

The most common B-cell lymphoma is EMZL, as previously reported, followed by DLBCL, FL, and MCL [13,14,15,16,17,18]. The results of this study were consistent with those reported in incidence studies of the ocular adnexal region, in which EMZLs are indeed found to be the most frequent lymphoma subtype. Among the less frequent subtypes, SLL and LL are usually observed [2,19]. The age distribution varies among these different subtypes, but, in general, orbital lymphoma primarily affects elderly patients. This applies especially to B-cell lymphomas, with the patients presenting this condition indeed being older than 50 years. This finding is also in line with other incidence studies of the ocular adnexa in which these four subtypes frequently occur in the elderly [11]. Our results are in line with this knowledge, given that most of our patients were elderly people (median age: 64 years). Furthermore, patients with MCL tended to be slightly older than patients with EMZL and DLBCL, also confirming previous reports [20].

From a clinical perspective, a variety of symptoms are frequently reported by patients with orbital lymphoma. Usually, the most common symptom of this condition is proptosis, especially when B-cell lymphomas are observed. Therefore, patients who present with proptosis, particularly when it is unilateral, should be referred to a tertiary referral center in order to be adequately assessed in a timely manner. Other commonly reported symptoms observed in these patients are limited eye motility, swelling, pain, changes in visual acuity, and diplopia, along with systemic symptoms such as fever, night sweats, or weight loss [21]. Our results are in line with this knowledge, as proptosis, swelling, and ptosis were the most reported symptoms.

After the identification of an orbital lesion that is highly suggestive of malignancy, performing a biopsy is a crucial step in the diagnostic workup of these patients, as this is able to confirm the diagnosis and be used for subtype classification. The biopsy should be performed as an open biopsy, as fine-needle aspiration is generally insufficient [22]. Further staging must be based on a full systemic workup, including a full-body PET-CT. One study [23] emphasized the role of this imaging technique for diagnosing simultaneous systemic presentations of orbital lymphoproliferative disease, confirming that PET-CT has higher sensitivity than CT alone in detecting systemic diseases [24,25,26]. This might have a significant role in the clinical management of these patients, as confirmed by our results, where subjects with PET-CT positivity presented lower survival rates compared to those with negative imaging. Furthermore, and similarly, as highlighted by our results, a bone marrow biopsy should also be performed in these patients [27]. Nonetheless, in the context of imaging, an unequivocable central role is played by MRI. This technique, given its inherent and intrinsic high-contrast resolution [28], can be used to successfully achieve diagnosis in a broad range of different clinical scenarios in either brain or head and neck imaging [29,30,31,32], including the field of orbital oncology. As advanced imaging techniques become widely popular in the scientific community [33], MRI (with all its conventional and advanced related sequences currently available) must be considered mandatory in all workups of patients with suspected OAL.

In this study, patients were staged according to the size and extent of their primary tumor (T), the involvement of local lymph nodes (N), and the presence or absence of metastatic spreading (M). The American Joint Committee on Cancer (AJCC) has created a staging system for ocular adnexal lymphoma (OAL) based on the TNM staging system. This system considers the site and extent of the tumor, and it has been used to classify OAL into different stages of malignancy and prognosis [34]. The TNM staging system includes several staging categories, resulting in a more even distribution of stages [1]. All the cases in this study were staged using the AJCC system, with the majority of patients that were classified belonging to a T2N0M0 stage. A recently published review by the American Academy of Ophthalmology on the treatment of OAL found that RTx has a very good effect on local control, disease-free survival, and overall survival in patients with EMZL [35]. Usually, RTx is chosen in patients with low-grade and localized (stage IE) primary lymphomas, showing a good effect on local control [35]. Furthermore, the use of RTx favorably altered the illness trajectory for OAL patients, with a particular reference to the improvement of linear accelerators able to progressively reduce the dose for marginal types from 40 Gy to 20 Gy and, in recent years, even less than 5 Gy [36], achieving an effective control of the disease and significantly reducing side effects. Regarding the therapeutic regimen, in our study, 63.8% of patients with EMZL received EBRT, a result in line with the current literature [37,38] showing the advantages of the use of EBRT in most cases of primary OAL, especially for low-grade lymphomas such as EMZL. Therefore, RTx in combination with chemotherapy may be used in patients with a systemic spread of cancer. Indeed, we found that chemotherapy or combined therapy regimens were the treatment modalities of choice for EMZL staged with T3N1M1 and DLBCL. Those with DLBCL received combined therapy regimens in 30% of cases. The histological subtype seems to be the most important prognostic factor for orbital lymphoma, and, overall, low-grade lymphoma subtypes have a better prognosis compared to high-grade lymphomas. In different studies evaluating the association between lymphoma subtype and survival in patients with OAL, histological subtype has been found to be the main outcome predictor [39,40,41]. It should be noted that these studies included patients with lymphoma of the ocular adnexa and not just lymphoma of the orbit. In accordance with the literature, in this study, we found that marginal lymphoma, or low-grade lymphomas, showed a higher survival rate compared to all other histological subtypes (*p* < 0.05).

## 5. Conclusions

Our investigation highlights the pivotal role of combined diagnostics in elucidating prognoses in patients affected by OALs. In particular, our findings suggest that in the EMZL subtype group, the presence of low-FDG avidity on PET-CT imaging and positive bone marrow involvement can significantly affect long-term prognosis. Taken together, our findings can provide researchers and clinicians with a relevant perspective into disease trajectory modelling and overall prognostic stratification. The potential integration of this combined diagnostics into prognostic clinical assessment models might offer meaningful insights into disease trajectory identification, treatment efficacy, and, therefore, prognoses for OAL patients overall.

## Figures and Tables

**Figure 1 cancers-16-02252-f001:**
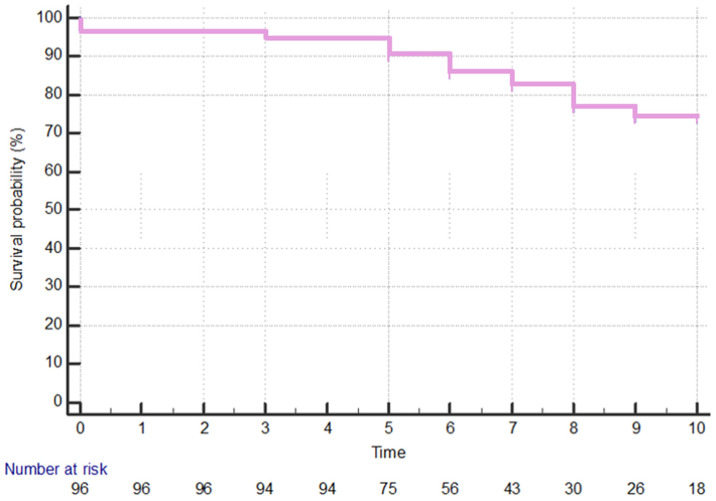
All lymphoma long-term cancer-specific survival. On the *x*-axis, the time, expressed in years, is reported, while on the *y*-axis, the percentage of survival probability is shown.

**Figure 2 cancers-16-02252-f002:**
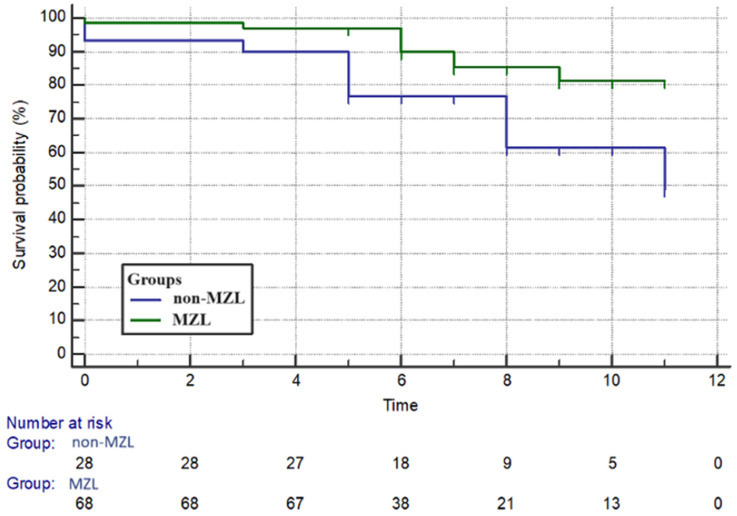
Survival rate—marginal zone lymphoma vs. other lymphomas. MZL = marginal zone lymphoma; non-MZL = lymphomas other than marginal zone. On the *x*-axis, the time, expressed in years, is reported, while on the *y*-axis, the percentage of survival probability is shown.

**Figure 3 cancers-16-02252-f003:**
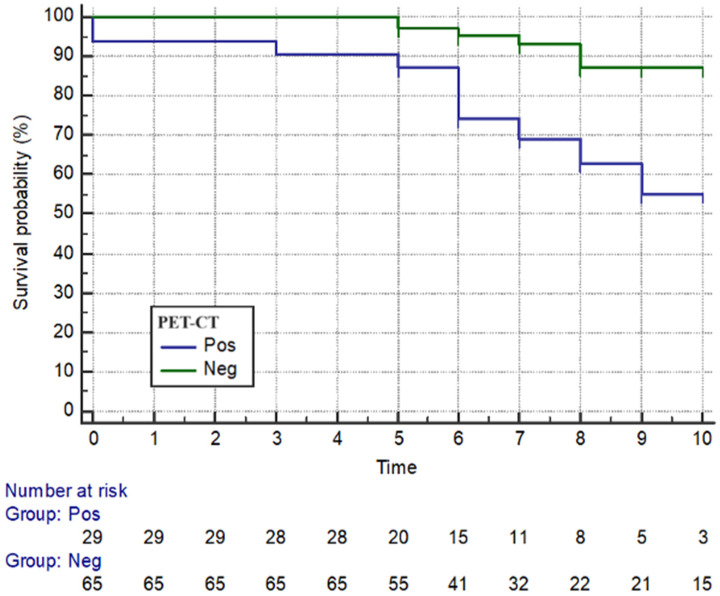
Survival rate—FDG-avid vs. non-avid lymphomas. Neg = Negative, without increased FDG uptake; PET-CT = Positron Emission Tomography–Computed Tomography; Pos = Positive, with an increased FDG uptake. On the *x*-axis, the time, expressed in years, is reported, while on the *y*-axis, the percentage of survival probability is shown.

**Figure 4 cancers-16-02252-f004:**
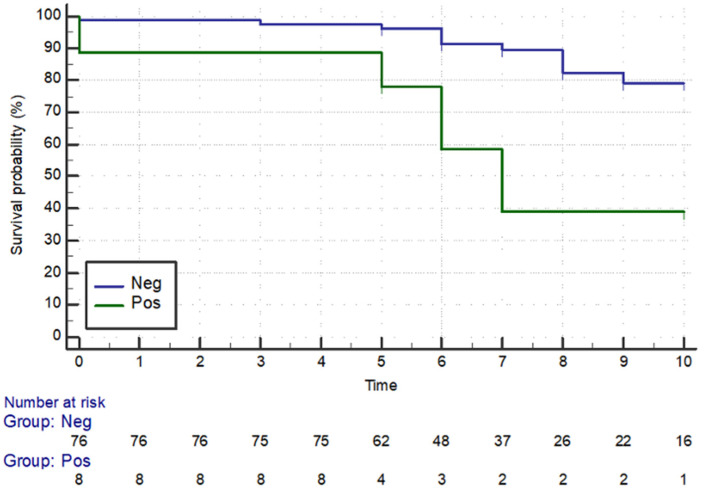
Survival rate—positive vs. negative bone marrow biopsy. Neg = Negative; Pos = Positive. On the *x*-axis, the time, expressed in years, is reported, while on the *y*-axis, the percentage of survival probability is shown.

**Figure 5 cancers-16-02252-f005:**
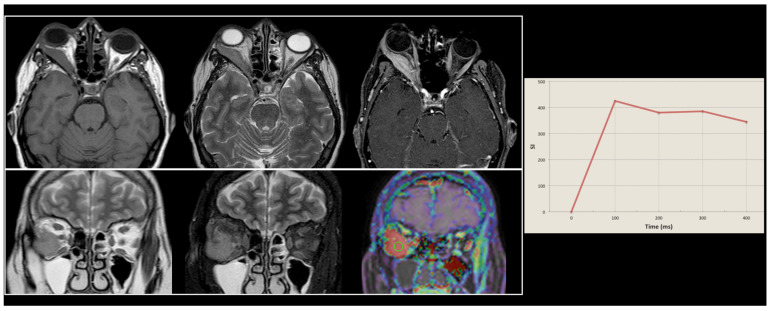
An example of orbital lymphoma. In the upper row, axial T1-weighted, T2-weigthed, and post-gadolinium T1-weighted images (from **left** to **right**), showing the presence of an expansive lesion adhering to the lateral wall of the right orbital cavity, determining a third-degree exophthalmos, showing a hypointense signal in T1w and T2w and intense enhancement in the late phases after Gadolinium administration. In the lower row, coronal T2w and STIR-T2w sequences (the two first images from the **left**) show the adhesion to the orbit floor and the displacement of the optic nerve, while the last image (**right**) shows a fusion image indicating a high degree of enhancement, suggestive of an aggressive lesion. In the right part of the image, a graph shows the rapid increase in signal intensity (SI, *y*-axis) over time (*x*-axis), which is highly suggestive of orbital lymphoma.

**Table 1 cancers-16-02252-t001:** Clinical and demographic data of each histopathologic subtype of OAL.

	EMZL	SLL	DLBCL	MCL	FL	LL
No. of patients	69	11	10	6	2	1
Gender (M:F)	36:33	5:6	7:3	5:1	0:2	0:1
Median Age	66	66	64.5	73	75.5	57
Location						
-orbit	45	10	8	2	2	1
-lacrimal gland	17	0	2	3	0	0
-eyelid	2	0	0	0	0	0
-conjunctiva	5	1	0	1	0	0
Laterality (Unilateral:Bilateral)	66:3	10:1	10:0	4:2	2:0	0:1
Clinical features at diagnosis						
-mass	19	3	4	1	1	1
-swelling	5	3	2	2	0	0
-ptosis	19	1	2	3	1	0
-epiphora	3	3	1	0	1	0
-exophtalmos	22	6	2	2	2	0
-diplopia	4	2	1	0	0	0
-conjunctival chemosis	5	2	0	1	0	0
-trigeminal paraesthesia	0	0	1	0	0	0

EMZL = extranodal marginal zone B-cell lymphoma; SLL = small lymphocytic lymphoma; DLBCL = diffuse large B-cell lymphoma; MCL = mantle cell lymphoma; FL = follicular lymphoma; LL = lymphoplasmacytic lymphoma.

**Table 2 cancers-16-02252-t002:** Number of cases treated with different interventions, stratified by diagnosis.

	RTx	CTx	RTx + CTx	No Treatment	CTx + Rituximab
EMZL	44	7	12	1	5
SLL	7	0	3	1	0
DLBCL	4	1	3	2	0
MCL	3	2	1	0	0
FL	0	2	0	0	0
LL	0	0	0	1	0

CTx = chemotherapy; DLBCL = diffuse large B-cell lymphoma; EMZL = extranodal marginal zone B-cell lymphoma; FL = follicular lymphoma; LL = lymphoplasmacytic lymphoma; MCL = mantle cell lymphoma; SLL = small lymphocytic lymphoma; RTx = radiotherapy.

## Data Availability

The raw data presented in this study are not available due to privacy restrictions; derived data can be requested from the corresponding author upon reasonable request.

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
