# Peer review of "Long-Term Survival and Factors Associated with Increased Mortality in Patients with Ocular Adnexal Lymphomas"

_cancers, 2024, doi:10.3390/cancers16122252_

Round 1

Reviewer 1 Report (Previous Reviewer 1)

Comments and Suggestions for Authors

As per the queries addressed by the reviewers I believe, authors have done a good job in addressing them. Accordingly the manuscript can be published with the incorporated changes.

Author Response

We thank the Referee for the positive review of our Work.

Reviewer 2 Report (Previous Reviewer 2)

Comments and Suggestions for Authors

The manuscript has significantly improved by the changes made by the authors. I have few comments:

- Please check all abbreviations, e.g. PET-TC is twice used. It should be CT?

- please unify style, e.g. p=... vs p =, i.e. with vs. without space

- I still can´t see p values in Tables 1 and 2

- Table 2: it is correct that none of the patients who underwent CTx + RTx received rituximab?

- please check references carefully, e.g. Ref. N. 15 deals with recurrent middle ear cholesteatoma and N. 32 with CMV encephalitis.

- Reference to Fig. 5 is missing in the text

- minor linguistic proof reading is required. I suggest to go through the paper by a native speaker

Comments on the Quality of English Language

please see above

Author Response

Reviewer 2

1) Please check all abbreviations, e.g. PET-TC is twice used. It should be CT?

We apologize for the clerical error, and we have modified the Manuscript accordingly.

2) please unify style, e.g. p=... vs p =, i.e. with vs. without space

We thank the Referee for the valuable suggestion, and in the revised version of the Manuscript we have modified the text accordingly (“p = ” in all cases”. Furthermore, we have decided to leave only one decimal after all p values, to further unify the style and increase readability of the Manuscript.

3) I still can´t see p values in Tables 1 and 2

We thank the Referee for giving us the opportunity to clarify this point. Indeed, p values in Tables 1 and 2 are not present because in these Tables are reported descriptive statistics only, without a direct comparison of these data (and therefore the corresponding expected p-values). We hope this clarifies.

4) Table 2: it is correct that none of the patients who underwent CTx + RTx received rituximab?

We confirm that the data in Table 2 are correct.

5) please check references carefully, e.g. Ref. N. 15 deals with recurrent middle ear cholesteatoma and N. 32 with CMV encephalitis.

We apologize for the clerical error. References are now placed in the correct context, along with other additional references to even further corroborate our Discussion.

6) Reference to Fig. 5 is missing in the text

A reference to Figure 5 in the main text was already present in the previous version of the Manuscript on Page 7, Line 172.

7) minor linguistic proof reading is required. I suggest to go through the paper by a native speaker

We thank the Referee for the valuable suggestion. The Manuscript has been now widely revised by an experienced Researcher with more than 10 years of experience in the field and Author of more than 130 works on indexed peer-reviewed Journals. We think that this thorough linguistic proof reading has significantly increased the overall quality of the Manuscript.

This manuscript is a resubmission of an earlier submission. The following is a list of the peer review reports and author responses from that submission.

Round 1

Reviewer 1 Report

Comments and Suggestions for Authors

In this current review “Ten-Year Survival and Factors Associated with Increased Mortality in Patients with Ocular Adnexal Lymphomas authors mention about the detailed study of ocular lymphomas which can be used for further prognosis. There are certain points in the study which are unclear, so the study can be published only if these points are addressed.

1)Authors mention about the ten-year analysis survival though the study mentions the data from 1 January 2000 through 31 December 2015

2)Full form of AJCC TNM classification system

3)In figure 1 the legend doesn’t specify about the entire details about x and y axis, why the x label axis is mentioned on the top of the graph?  

4) In order to avoid confusion in the Figure 1 authors should color the curve in a different color as blue color in rest of the figures denotes Non_MZL lymphoma.

5)In Figure 2 authors mentions about the time till 12 years, shouldn’t the authors follow uniformity having 10 years in this too.

6) In Figure 5 mentioning the conclusion independently is not the appropriate way to address the information.

7)In Figure 5, the legend is bit confusing as it mentions the texts but doesn’t label any of the information in the figure, they should make different panels in the same figure labelling the required information.

8)In Figure 5, the graph is blurred and is not clear, it should be mentioned separately, with the clear labels and legends.

9) RT is Radiotherapy should be mentioned in the text.

10) The details about the Figure 5 have no mentions in the results which is unusual and questions the importance of the study. Authors need to explain what Fig. 5 signifies in the results.

11) Authors can tell the readers about conjunctival lymphoma and intraocular lymphoma for the ease of understanding the study and does they have a correlation.

12)It is already known that RT plays an important role in prognosis of lymphomas, authors should try to explain in few lines that knowing about disease trajectory can be beneficial in understanding the future prognosis. 

13)Authors should have uniformity in writing references.

Reviewer 2 Report

Comments and Suggestions for Authors

The authors report on 99 patients with ocular lymphoma. Ocular lymphoma is quite rare, thus the data are of potential importance. However, parts of the manuscript (especially the discussion) are quite poorly written and it is difficult to go through them.

Ø  Why have patients only until the year 2015 been included in this study and not more recent ones?

Ø  Why have only 5 patients received rituximab? This study was mainly done in the rituximab era and this agent is typically used for systemic treatment of B cell lymphoma.

Ø  Several parts of the manuscript are far too long and redundant.

Ø  The reference list is very diffuse, e.g. references N. 32 and 33 are missing (skipped in the list).

Ø  Several statements in the text does not fit to the reference list, e.g. reference N. 13.

Ø  The authors state that “One study emphasized the role of PET-CT for diagnosing simultaneous systemic presentations of orbital lymphoproliferative disease and suggested that PET has higher sensitivity than CT in detecting systemic disease [27, 28]. However, Ref. N. 27 and 28 do not deal with PET CT scan.

Ø  Table 1. Symptoms are unclear. Likely, several patients might have had more than one of these symptoms, thus the number should be higher than the total number of patients included.

Ø  In the discussion it is unclear if several statements refer to findings made in the literature or results from the present study.

Ø  P values are missing in table 1.

Ø  This statement is unclear: “Indeed all our enrolled patients with PET-CT positivity presented a lower survival rate compared to negative ones.

Ø  It is not correct that all lymphoma patients undergoing PET CT need also bone marrow analysis.

Ø  Figure N. 5 appears before Figures N1-4 in the text.

Ø  Abbreviations are not used and explained uniformly.

Comments on the Quality of English Language

/